# New Insights into Iron Deficiency Anemia in Children: A Practical Review

**DOI:** 10.3390/metabo12040289

**Published:** 2022-03-25

**Authors:** Carla Moscheo, Maria Licciardello, Piera Samperi, Milena La Spina, Andrea Di Cataldo, Giovanna Russo

**Affiliations:** 1Pediatric Unit, Azienda Ospedaliero Universitaria Policlinico “Rodolico-San Marco”, viale Carlo Azeglio Ciampi n.1, 95121 Catania, Italy; cmoscheo@gmail.com; 2Pediatric Onco-Hematology Unit, Azienda Ospedaliero Universitaria Policlinico “Rodolico-San Marco”, via Santa Sofia n.78, 95123 Catania, Italy; marinella72@icloud.com (M.L.); psamperi@unict.it (P.S.); mlaspina@unict.it (M.L.S.); 3Department of Clinical and Experimental Medicine, University of Catania, via Santa Sofia 89, 95123 Catania, Italy

**Keywords:** anemia, iron-deficiency, pediatric, iron therapy, bis-glycinate iron, liposomal iron, iron deficiency prevention

## Abstract

Iron deficiency anemia (IDA) is the most frequent hematological disorder in children, with an incidence in industrialized countries of 20.1% between 0 and 4 years of age and 5.9% between 5 and 14 years (39 and 48.1% in developing countries). Although IDA has been recognized for a long time, there are still uncovered issues and room for improving the management of this condition. New frontiers regarding its diagnosis and therapeutic options emerge every day; recently, innovative formulations of iron have been launched, both for oral and parenteral administration, with the aim of offering treatment schedules with higher efficacy and lower toxicity. As a matter of fact, glycinate and liposomal preparations, while maintaining a satisfying efficacy profile, have significantly fewer side effects, in comparison to the traditional elemental iron salts; parenteral iron, usually considered a second-choice therapy reserved to selected cases, may evolve further, as a consequence of the production of molecules with an interesting clinical profile such as ferrocarboxymaltose, which is already available for adolescents aged >14 years. The present article reports the clinically relevant latest insights regarding IDA in children and offers a practical guide to help pediatricians, particularly to choose the most appropriate prevention and therapy strategies.

## 1. Definition and Etiopathogenesis

Iron deficiency anemia (IDA) is the most frequent hematological disorder of childhood and adolescence and the most common form of anemia, with an incidence in industrialized countries of 20.1% between 0 and 4 years of age and 5.9% between 5 and 14 years (39 and 48.1% in developing countries) [1]. It is a hypochromic and microcytic anemia characterized by Hb values below the normal range for sex and age, reduced MCV, and MCH.

Iron is an essential nutrient for the development of the fetus, infant, and child [2]. The body’s iron content is dependent on its intake and absorption with nutrition. The homeostasis of this nutrient is determined by the balance between its uptake and release from the cells where it is stored and recycled [3,4]. Iron is released into the circulation, where it is carried by the plasma protein transferrin, into the duodenum by enterocytes that absorb dietary iron, and by macrophages which recycle senescent erythrocytes and liver reserves [3]. If iron levels in the body are inadequate, its intestinal absorption is enhanced [5]; in case of excess, it is stored in the enterocytes as ferritin and the liver, spleen, and bone marrow as hemosiderin [6]. The release of free iron ions in the plasma, essential for the maintenance of its homeostasis, is mediated by ferroportin, whose expression is subordinated to the activity of hepcidin [7].

## 2. Risk Factors and Predisposing Conditions

The pathogenesis of iron deficiency anemia is complex. It can be caused by a reduced dietary intake of iron or increased need, acute or chronic blood loss, and intestinal malabsorption of iron (Table 1). During childhood, IDA has two peaks of incidence in infancy and adolescence, when there is a discrepancy between food intake and high need of iron depending on the remarkable growth speed.

There are several paraphysiological conditions that during the different stages of growth can favor the onset of IDA in the absence of an underlying disease: in the first months of life, the need for iron is satisfied by the iron deposits acquired through the transplacental exchange during intrauterine life and by maternal breastfeeding. After the first months of life, the increased erythropoiesis and the depletion of prenatal iron stores increase the demand for iron intake, which is no longer satisfied by breast milk, making it necessary to promptly and adequately wean in order to ensure the inclusion of iron-rich foods. The prolongation of exclusive breastfeeding may be related to the high cost of foods containing heme iron, as frequently occurs in developing countries, or to the erroneous opinion that milk is healthful nutritious, without considering that the prevalent intake of milk prevents the consumption of other iron-rich foods. In fact, nutritional deficiency is still the most common cause of iron deficiency even in industrialized countries: a recent Italian survey showed that the most frequent cause of iron deficiency is exclusive breastfeeding beyond 6–12 months of life and identified foreign-born preschool children as the population at highest risk of iron deficiency [8]. Family pediatricians must closely monitor the eating habit of their patients as dietary errors can depend on cultural factors, trends, or inattention. The need for iron also increases in another phase characterized by a rapid increase in body weight: adolescence. During this age, other contributing causes can compromise iron supply: low intake with the diet, often inadequate, consumption of food products that reduce its absorption, malnutrition, obesity, and iron deficiency associated with sport, and, for females, the loss of iron during the menstrual cycle. 

Among the causes of IDA (summarized in Table 1), the following conditions deserve particular attention from the Pediatricians: Prematurity: preterm infants are at high risk of developing IDA, both because 80% of iron is acquired by the fetus during the last trimester of pregnancy and for the greater speed of growth in the first months of life.Neuromotor disorders: IDA is common in these children, mainly due to swallowing impairment that requires the use of nutrition by gavage and liquid/semi-liquid food with possible exclusion from the diet of various categories of nutrients. In addition, the presence of other disorders, such as gastroesophageal reflux, is frequent, which, being complicated by the onset of esophagitis, can be the cause of chronic bleeding.Diseases of the gastrointestinal tract: iron malabsorption, as occurs in the presence of celiac disease, *Helicobacter pylori* (*H. pylori*) infection, chronic inflammatory bowel disease, pernicious anemia, and prolonged use of drugs as pump inhibitors, and conditions determining chronic blood loss (intolerance to cow’s milk proteins, Meckel’s diverticulum, hiatus hernia, intestinal parasitosis) are causes of IDA refractory to oral treatment. In particular, IDA during *H. pylori* infection may depend on an impediment to intestinal iron absorption and possible bleeding. Several studies showed improvement of IDA following antibiotic eradication therapy in *H. pylori* infection [9,10]. Serology for celiac disease and search for *H. pylori* should therefore be included in the early phase of the diagnostic workup in cases of IDA over two years of age, particularly in case of refractoriness to iron treatment (Figure 1). IDA is also the most frequent extra-intestinal complication of chronic intestinal inflammatory diseases being attributable to an increase in the body’s energy requirements, insufficient dietary intake, reduced enteric absorption, and intestinal blood loss.Among the extra-intestinal blood losses, it is worth mentioning the irregularities of the menstrual cycle, frequent in adolescents, and, rarely, secondary to Von Willebrand disease, a constitutional coagulopathy characterized by an absent to slight hemorrhagic diathesis, which may remain undiagnosed up to the menarche [11].Refractory IDA is defined by lack of response to oral treatment, after careful exclusion of all possible factors leading to poor adherence to therapy, such as insufficient dosage, the timing of administration, type of iron administered, duration of therapy, presence of inflammation or infection. Usually, these are secondary forms, and the identification of the underlying cause may be challenging (Figure 1).

Among unexplained IDA, IRIDA (iron-refractory iron deficiency anemia) should be considered. It has a genetic basis linked to the TMPRSS6 gene mutation. Patients carrying this mutation have high levels of hepcidin, a hormone that negatively regulates the absorption of iron in the gastrointestinal tract and explains the lack of responsiveness to oral therapy. Probably this gene encodes for a protease responsible for the downregulation of hepcidin synthesis [12,13,14,15,16].

## 3. Clinical Features of Iron Deficiency

Mild to moderate iron deficiency, not associated with anemia, can be asymptomatic or lead to fatigue and/or poor tolerance to exercise [17]. Typical clinical presentation of moderate to severe iron deficiency includes the usual signs of anemia, such as pallor and fatigue. More specifically, restless leg syndrome, mucus trophic lesions (stomatitis, glossitis), pica, frequent infections, mood, and behavior disorders with a decline in school performance may occur. Sideropenia during infancy has been related to permanent neurocognitive impairments, reduced learning capacity, and altered motor ability [18,19]. 

## 4. Diagnosis

Laboratory reveals microcytic-hypochromic anemia (reduced Hb, MCV, MCH, elevated RDW) with reduced reticulocyte count. Low ferritin, sideremia, a saturation of transferrin, and high unsaturated serum transferrin are noted. Usually, ferritin is the reference test for assessing the state of iron deposit; however, the lower limits vary according to age and sex. False normal to high levels of ferritin are typical of the inflammatory or infectious state. Therefore, all these issues impair the usefulness of ferritin as an accurate marker of iron status. Free erythrocyte protoporphyrin increases in iron deficiency anemia, but it is a non-specific parameter [20].

## 5. Newly Introduced Markers in the Diagnostic Framework of Iron Deficiency Anemia

More recent diagnostic investigations include serum soluble transferrin receptor (sTFRC) assay, which is increased in the presence of iron deficiency anemia. Iron deficiency induces the expression of the transferrin membrane receptor, which, in turn, determines the concentration of serum soluble receptor (sTFRC). The concentration of sTFRC is not influenced by inflammatory or infectious processes and is therefore very useful for diagnostic purposes [20]. The ratio of sTFRC to ferritin, which is low in iron deficiency anemia and high in chronic disease anemia, is useful in the differential diagnosis between the two conditions. 

Another marker of iron metabolism is serum hepcidin, whose levels are significantly reduced in IDA (since iron absorption should not be inhibited in this context), elevated in chronic disease anemia and in obesity (in response to inflammatory stimuli, mediated by interleukin-6, hepcidin is released by hepatocytes), and inadequately normal or high in genetically determined iron deficiency anemia (IRIDA). The increase in the serum hepcidin value indicates even a non-responsiveness to oral iron therapy in adults [21].

Hepcidin is a peptide produced in the liver that plays a crucial role in the metabolism of iron by regulating its cellular transport [22]. The cellular content of iron is dependent on the amount present in the circulation carried by the plasma protein transferrin. It is released into the circulation by enterocytes which in the duodenum absorb iron introduced with food, macrophages that acquire it from senescent erythrocytes, and liver reserves. The release of iron into the circulation by these cellular elements is mediated by the only known iron exporter, ferroportin (FPN). FPN is a protein molecule expressed at the duodenal, hepatic, splenic, and placental levels [23,24,25]. Hepcidin, by binding to FPN and inducing internalization and degradation, inhibits the release of iron from cells [22,26,27], negatively impacting the absorption of iron from the duodenum and its release from the macrophages of the liver and spleen [28,29] (Figure 2).

Regretfully, measurement of sTFRC and hepcidin are not commonly offered and are still expensive, and are currently used almost exclusively for research purposes. 

Table 2 summarizes the reference values of the most commonly used markers in the diagnostic setting of iron deficiency anemia [30,31]. It is fundamental to point out that the determination of the main indicative markers of anemia can make use of various diagnostic methods for the acquisition of blood samples (such as venous and capillary samples) and that, in particular for the determination of hemoglobinemia values, there are also portable haemoglobinometers, which are very convenient to use with children since they allow to detect anemia more quickly, easily and also less invasively [32]. 

## 6. Oral Iron Therapy

IDA therapy usually relies on oral administration of iron; it is the usual first choice due to its excellent efficacy/safety/cost profile [33]. The most used preparations are ferrous sulfate or gluconate due to their high intestinal absorption (10–15%). 

However, despite the well-known efficacy, gastrointestinal adverse effects, such as abdominal pain, dyspepsia, nausea, vomiting, diarrhea, or constipation, are a major concern of oral ferrous salts, affecting up to 32% of patients [34]. A possible consequence is a low acceptability, scarce adherence to therapy and continuation of IDA. Although ferrous compounds should be administered far away from meals to ensure better absorption, the side effects can be limited if iron is administered on a full stomach. 

There are no reports of clinical trials comparing various oral iron preparations, their dosage, and duration [35]. Therefore, despite the usual dosage in textbooks being 3 to 6 mg/kg/day of iron, twice or thrice in a day, evidence is gathering on the use of lower doses, with the aim of having equal or even superior efficacy and fewer side effects, both in adults [36,37,38,39] and children [40,41,42]. A possible explanation is the role of hepcidin, a hormone that negatively regulates intestinal absorption of iron and its release by macrophages. The synthesis and hepatic release of hepcidin are regulated by an increase in iron serum and intrahepatic iron deposits (Figure 2). A transient rise in iron and ferritin values, as occurs immediately after oral iron administration, causes a release of hepcidin, reduced absorption of iron in the intestine, and an increase in potential iatrogenic side effects. This reduced intestinal absorption, consequent to oral martial therapy, has been given the name of the “hepcidin effect”. Current therapeutic schedules that provide from 1 to 3 daily administrations would favor the persistence of this effect, with a greater incidence of gastrointestinal side effects and reduced adherence to treatment. Given the demonstration that the hepcidin effect has a duration of approximately 48 h, this could be avoided by using a therapeutic regimen that provides oral administration of iron once every 48 h [37]. 

The search for a lower dosage of oral iron also derives from concerns on a possible negative impact of excess oral iron on the gut microbiota, favoring the nourishment of pathogenic bacteria [43]. Furthermore, excess iron, especially in iron-rich foods such as red meat and iron-enriched food, has been related to the occurrence of colorectal carcinoma; possible pathogenetic mechanisms include iron’s mutagenic effect due to ability to produce free-radicals and its nutritional support to cancer cells; there is evidence that iron fortification of staple foods is associated with increased incidence of colorectal carcinoma [44]; and, in general, there is a correlation between iron intake and colorectal carcinoma [45,46]. Even if there are no reports on a specific correlation between pharmacologic administration of iron and cancer, the epidemiologic reports on the possible role of iron in the onset and advance of cancer suggest caution to avoid excess iron consumption [47,48,49].

Other strategies to limit side effects include formulations such as bis-glycinate iron and liposomal iron, which have higher bioavailability and fewer gastrointestinal adverse effects. In iron bis-glycinate, the amino acid glycine chelates iron, forming a chemically inert compound that is absorbed in the intestinal mucosa, a mechanism that allows it to be absorbed 3–4 times higher than iron sulfate. Bis-glycinate iron proved to be effective as an addition to fortifying food [50], to supplement pregnant women [51], and has a superior bioavailability [52]. On top of that, it is reported to cause fewer side effects when compared with elemental iron [53,54]. Its safety profile also seems adequate. 

Liposomal iron, a preparation of ferric pyrophosphate transported within a phospholipid membrane, is absorbed by the intestinal M cells, then poured out on the lymphatic system, transferred to the liver, and finally released. This mechanism determines its greater absorption compared to ferrous salts and avoids chemical interaction with the gastrointestinal wall, therefore limiting its side effects. Liposomal iron has been proved to be both efficacious and safe in chronic kidney disease patients [55]. A very recent observational study reports that these preparations are widely used in clinical practice with satisfactory results [56].

The response to oral iron therapy is profiled with the onset of reticulocytosis after 72–96 h from its onset and an increase in the Hb rate starting from the 4th day. This response can be very useful in clinical practice, especially if oral therapy is started in a patient with severe anemia. It promptly identifies children who are responsive to therapy and allows them to wait for the hemoglobin response, which can be appreciated no earlier than 7–10 days. The hemoglobin normalizes after the first month, and the iron stores are reconstituted between the first and the third month. Therefore the treatment must be continued, for at least three months, after the normalization of the Hb values, in order to restore iron deposits [56]. Table 3 lists currently available treatment options. 

## 7. Parenteral Iron Therapy

Parenteral therapy is indicated in case of intolerance or refractoriness to oral therapy, inability to swallow iron preparations, chronic malabsorption; severe symptomatic anemia is not in itself an indication for parenteral iron, since it has been reported a prompt and favorable response to oral iron also when circulating hemoglobin is low [41,42]; parenteral iron bypasses the intestinal absorption and is considered to determine a faster response than the oral route; it allows to calculate exactly the dose of iron necessary to achieve the normalization of the hemoglobin level and the restoration of iron reserves and has fewer gastrointestinal side effects. Its limits are the greater impact on the child (venipuncture, hospitalization) and the higher cost. Moreover, pediatric studies on its safety and efficacy are limited. High molecular weight iron dextran has been abandoned due to the high risk of anaphylactic reactions. Currently available iron compounds (Table 3) are those of the second generation, such as low molecular weight iron dextran, iron saccharate, and ferric gluconate. The latter two compounds are the most widely used, as they are safe and not associated with the risk of severe anaphylactic reactions [57]. A major concern is the need to divide the total amount of required iron in repeated infusions. The third-generation compounds, such as ferric carboxymaltose and iron isomaltose, have the advantage of being able to be administered in high doses in a single shot; only ferric carboxy-maltose has an indication for pediatric patients over the age of 14; it requires a lower dosage and a shorter infusion time than second-generation preparations. It should be noted that in patients undergoing iron therapy, the risk of developing iron overload, also burdened by a potential pro-infectious effect, must always be monitored. Regarding parenteral therapy-related adverse events, clinical studies show that severe adverse reactions are rare events and can be contained by close monitoring of patients and identification of at-risk populations. 

## 8. Prevention of Iron Deficiency

The content of iron in different foods varies greatly; Table 4 reports some examples of the amount of iron in various categories of foods, which shows that some nutrients, such as spinach and legumes, may be erroneously thought to be a better source of iron than cereals. 

The effect of the diet composition on iron homeostasis depends not only exclusively on the iron content but also on the amount that is absorbed and available. Iron is contained in food as heme and non-heme iron [58]. 

Heme iron, deriving from the breakdown of hemoglobin and myoglobin in meat and fish, has a higher bioavailability [59]. The bioavailability of iron is the amount of it that the body is able to absorb and use for bodily functions; the bioavailability of a nutrient depends on several factors, including its bioaccessibility, for example, the amount that can be released from the matrix during digestion of food and pass into the soluble fraction, becoming available for absorption by the body through the epithelial cells of the gastrointestinal mucosa [60]. The first step in making a nutrient bioavailable is to free it from the food matrix and transform it into a chemical form that can bind and enter between intestinal cells. The nutrients are made bioaccessible by the processes of chewing and enzymatic digestion of food. The small intestine is the main nutrient absorption site. Although the terms bioavailability and bioaccessibility are often used interchangeably, it is important to note that bioavailability includes bioaccessibility [61].

Non-heme iron contained in cereals, legumes, and vegetables has a much lower bioavailability than heme iron. Furthermore, the absorption of non-heme iron is influenced by the concomitance of other substances in the meal: phytates, tannates, and phosphates reduce the absorption of non-heme iron; others, such as ascorbic acid, facilitate it (Figure 3); recently also other nutrients have been reported to interfere with iron utilization, such as dietary vitamin A [62]. In order to find efficient strategies to alleviate iron deficiency, studies are underway to maximize the bioaccessibility of iron and other nutrients [63,64]. 

Prevention of IDA must be implemented already during pregnancy since maternal sideropenia affects its occurrence and possible sequelae during infancy [19]. In fact, it has been shown that perinatal iron deficiency can lead to delayed neurocognitive development, which seems to persist in childhood despite treatment with oral iron. It is considered crucial in the development of the sequelae attributable to IDA in the period of one thousand days, including gestation and the first two years of life. The World Health Organization (WHO) recommends supplementation of iron and folic acid in areas with a high prevalence of anemia in women of reproductive age and during gestation [1].

At birth, delayed clamping of the umbilical cord is a measure to increase the values of hemoglobin and ferritin and reduce the risk of IDA in the preterm infant [65].It is important to promote breastfeeding at birth, or the use of iron-enriched formulas for the first year of life, avoiding cow’s milk that contains iron with low bioavailability.In the neonatal period, prophylactic oral iron supplementation is used in preterm and low birth weight births. The American Academy of Pediatrics recommends 2 mg/kg/day of elemental iron, starting from one month up to one year of age [18].When exclusive breastfeeding is prolonged after four months, the American Academy of Pediatrics recommends a supplementation of elemental iron, 1 mg/kg/day, until iron-rich foods are introduced [18].At the time of weaning, it is advisable to include foods containing iron with high bioavailability and substances that favor its absorption; it is worth mentioning that meat and fish not only contain heme iron, with high bioavailability but also favor the absorption of non-heme iron in the same meal (Figure 3); the consumption of foods for infants, specifically enriched in iron, is recommended; the “universal” strategy of fortifying food for the population with iron poses various problems of palatability and conservation and should be taken into consideration as a measure in developing countries [1]. A warning must be mentioned regarding the excessive use of milk in infants and toddlers: it is a common opinion that milk is healthful and nutritive, ignoring that a great intake of milk inhibits the inclusion of iron-rich foods in the diet [66]. Weaning is a crucial transitional phase, influenced by several factors: cultural, behavioral, social issues play an important role; parents and caregivers need to be guided through the process, and pediatricians should actively monitor that children’s nutrition is adequate, as needed. In industrialized countries, particular attention has to be devoted to newly arrived immigrants [8,56]The vegetarian diet is an increasingly popular diet. The American Dietetic Association and Dietitians of Canada state that a well-structured vegetarian diet is suitable for all ages [67]. Some studies show that vegetarian children in industrialized countries have an iron intake similar to omnivores, with a prevalent intake of iron characterized by a lower bioavailability and elements that implement its absorption. They also note that, although parameters such as hemoglobin, MCV, red blood cells are comparable in vegetarian and omnivorous children, in the former, there is a lower level of serum ferritin, which indicates lower reserves of this element [68]. There are no data in the literature that place a univocal indication for iron supplementation in the vegetarian population.

## 9. Conclusions

Iron deficiency anemia is the most diffuse anemia in childhood. Although much has already been learned about this, new frontiers in diagnosis and therapy emerge every day. The identification and characterization of the hepcidin molecule could help to better define the condition of sideropenia and monitor the response to treatment in the presence of certain etiopathogenetic moments, such as concomitant infection, anemia related to inflammation, and genetic iron deficiency anemia refractory to oral treatment (IRIDA). Concerning oral therapy, the use of glycinate and liposomal preparations should be further explored since preliminary data suggest their efficacy with a lower incidence of side effects compared to other formulations. Concerning parenteral therapy in the pediatric age, new developments are expected. Molecules such as ferrocarboxymaltose, deliverable in single administration with efficacy and mild/moderate adverse events, could be a valuable option, yet they are currently off-label under the age of 14. When considering dietary habits, the literature indicates the possibility of observing vegetarian-type diets with a good safety profile, always under careful monitoring by the pediatrician. Further studies are therefore needed to improve the knowledge and diagnostic-therapeutic interventions related to such a widespread disorder.

## Figures and Tables

**Figure 1 metabolites-12-00289-f001:**
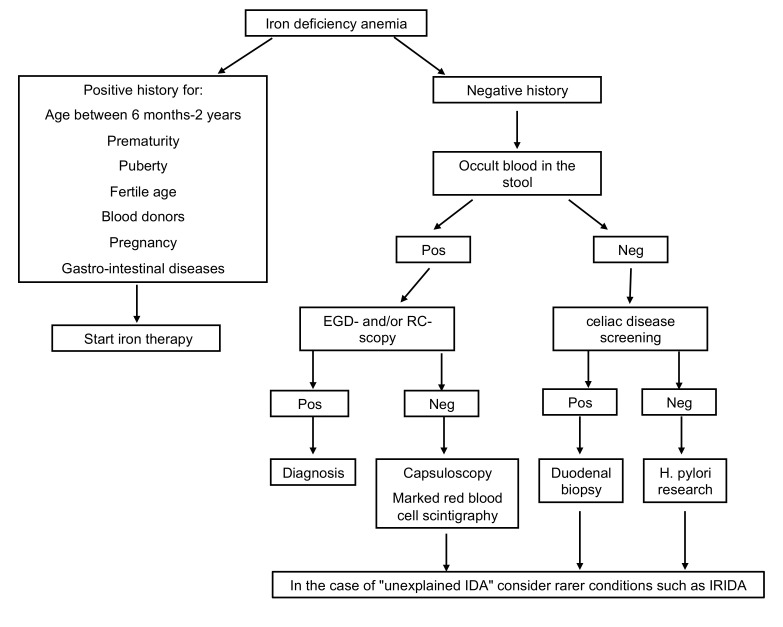
Diagnostic workup in case of IDA. EGD-scopy, esophagogastroduodenoscopy; RC-scopy, rectoscopy; Pos, positive; Neg, negative; *H.pylori*, *Helicobacter pylori*; IDA, iron deficiency anemia.

**Figure 2 metabolites-12-00289-f002:**
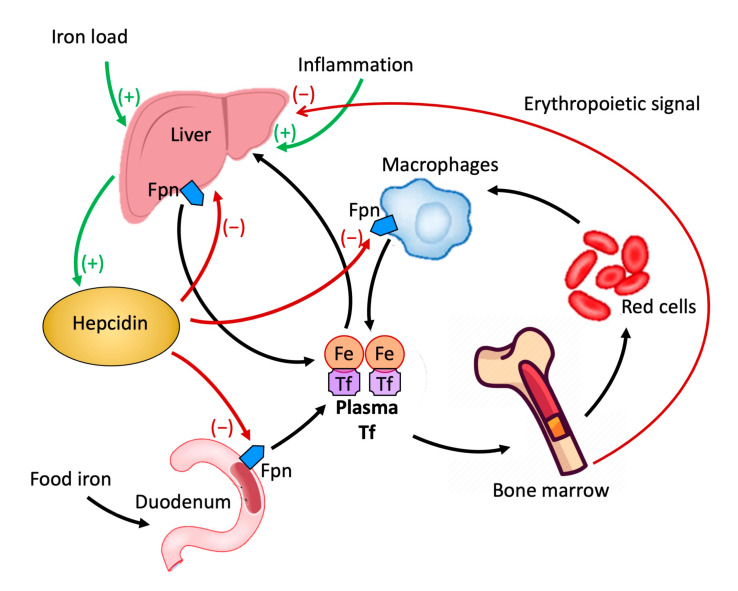
Role of hepcidin in the homeostasis of iron.

**Figure 3 metabolites-12-00289-f003:**
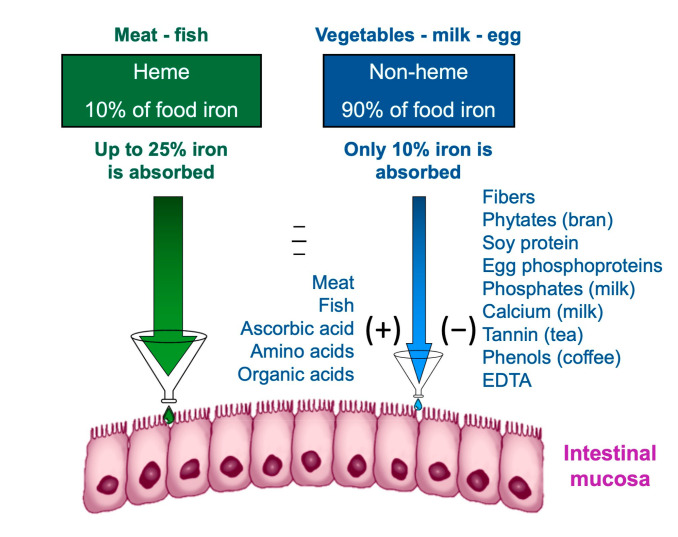
Intestinal iron absorption is influenced by diet composition.

**Table 1 metabolites-12-00289-t001:** Causes of iron deficiency anemia (IDA).

Mechanisms Leading to Iron Deficiency	Conditions
Decreased dietary iron supply	Prematurity
	Late weaning
	Vegetarian diet
	Swallowing disorders
Increased iron demands	Infancy
	Low birth weight
	Pubertal spurt growth
Reduced intestinal iron absorption	Celiac disease
	*Helicobacter pylori* infection
	Chronic autoimmune gastritis
	Use of protein-pump inhibitors
	Inflammatory bowel diseases
	IRIDA
Blood loss	Heavy and/or frequent menses
	Cow milk protein intolerance
	Meckel diverticulum
	Hiatal hernia
	Intestinal parassitosis
	Inflammatory bowel disease
	Bleeding diathesis

**Table 2 metabolites-12-00289-t002:** Cutoff points according to age for hematometric iron indicators.

Age	Hb g/dL	MCV fL	Serum Iron μg/dL	SF ng/mL	Tfs %	TIBC μg/dL	Hep nmol/L
Newborn	<13.2	<100	<63	<6	<30	>285	
6–24 months	<11.3	<68–72	<35	<6	<10	>434	1.1–7.3 (M) 0.9–7.5 (F)
2–6 years	<11.5	<75	<22	<6	<7	>441	1.0–3.3 (M) 1.1–3.8 (F)
6–12 years	<12	<77	<39	<10	<17	>508	0.9–3.4 (M) 0.9–3.9 (F)
12–18 years	<13 (M) <12 (F)	<78		<23 (M) <6 (F)	<6	>470 (M) >564 (F)	0.3–1.8 (M) 0.5–2.2 (F)

SF, serum ferritin; Tfs, transferrin saturation; TIBC, total iron-binding capacity; Hep, hepcidin; Hb, hemoglobin; MCV, mean corpuscular volume; M, male; F, female. Source: Nathan and Oski’s *Hematology of Infancy and Childhood*, Seventh edition, Elseviers, 2009.

**Table 3 metabolites-12-00289-t003:** Available iron formulations for pediatric prescription.

Iron Formulation	Recommended Dosage *	Benefit	Criticality	Note
Iron sulfate Iron gluconate	2–6 mg/kg/day	Standard treatment Good intestinal absorption Low cost	Gastro-enteric side effects in 15–32% of cases Unpleasant taste Drop preparation rarely available	A low dosage, i.e., 2 mg/Kg/day, has been proposed as a still efficacious and better-tolerated schedule
Iron glycinate	0.45 mg/kg/day	Good intestinal absorption Limited side effects Drop formulation available		
Liposomal iron	1.4 mg/kg/day	Excellent palatability No side effects Drop formulation available	Possible less prompt response to therapy	
I.v. iron gluconate	Total dose to be calculated based on initial Hb and weight	Effectiveness independent of gastro-enteric absorption Very low gastro-enteric side effects	Hospitalization required Multiple infusions	
I.v. carboxymaltose iron	Dose to be calculated based on initial Hb and weight	Effectiveness independent of gastro-enteric absorption Single administration	PHospitalization required	Indication for adolescents ≥ 14 years

* Dosage is based on available evidence and/or textbooks.

**Table 4 metabolites-12-00289-t004:** Iron content in food *.

Food Category	Food	Iron Content (mg/100 g of Food)
Heme iron		
Meat	Turkey, calf, bovine, horse	2–4
	Beef liver	8.8
	Beef spleen	42
Fish	Cod Sea bass	0.9 4.0
	Shrimp	2.6
Non-heme iron		
Egg	Whole egg	1.5
	Egg yolk	4.9
Cereals	Bread	2.5
	Pasta Rice	2.5 2.9
Legumes	Fresh legumes	2.3
	Dry legumes	6–8
Fruit	Fresh fruit Olive	0.4–0.5 1.6
	Dry fruit-nuts	2.1–2.6
Vegetables	Spinach, tomato, potato, artichokes, lettuce	0.4–1.3
Dairy products	Milk Yogurt Parmesan cheese	0.1–0.3 0.1 0.2

* Source: https://www.crea.gov.it/ (accessed on 25 February 2022).

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
