# Peer review of "New Insights into Iron Deficiency Anemia in Children: A Practical Review"

_metabolites, 2022, doi:10.3390/metabo12040289_

Round 1

Reviewer 1 Report

In places the English is somewhat idiosyncratic, for example: “alimentary” – change to “dietary” is more suitable.

The authors partially complied reviewers' requests. The content seems superficial to me, mainly in the definitions of bioavailability and bioaccessibility. English revision is necessary, perhaps it will make it clearer. Despite the difficulties, I believe that its publication, after rigorous review, will be useful.

Author Response

Question 1

In places the English is somewhat idiosyncratic, for example: “alimentary” – change to “dietary” is more suitable.

Answer 1.

“Alimentary” has been chaged to “dietary” in table 1 (see highlighted in the text)

English has been reviewed by a native English speaker and many corrections have been made

Question 2

The authors partially complied reviewers' requests. The content seems superficial to me, mainly in the definitions of bioavailability and bioaccessibility.

Answer 2.

Bioavailability and bioaccessibility have been explained deeply

Reviewer 2 Report

The authors have addressed my previous comments and those of the other reviewers, the manuscript is greatly improved and is now suitable for publication. One minor point is that ref 22 is quite old and there are many other excellent reviews covering the more recent advances in the field of the fpn-hepcidin axis.

Author Response

Question

One minor point is that ref 22 is quite old and there are many other excellent reviews covering the more recent advances in the field of the fpn-hepcidin axis.

Answer

Ref 22 has been replaced by a more recent paper

Reviewer 3 Report

Whilst I still think this is a fairly low level review, I do not object to its publication. 

Author Response

Thank you for your final comment

Round 2

Reviewer 1 Report

The authors responded to the reviewers' suggestions.

Congrats on the study.

This manuscript is a resubmission of an earlier submission. The following is a list of the peer review reports and author responses from that submission.

Round 1

Reviewer 1 Report

The study “New insights into iron deficiency anemia in children: a practical review practical guide to help pediatricians, particularly in the choice of the most appropriate prevention and therapy strategies” aimed to report the clinically relevant latest insights regarding IDA in children and offers a practical guide to help pediatricians, particularly in the choice of the most appropriate prevention and therapy strategies.

The idea of ​​the study is relevant and pertinent. However, the content at some points seemed superficial to me, and the more complex information was left out due to the lack of a previous theoretical basis. I suggest that some basic concepts be included in the text in a summarized way, among them: absorption and use of iron by the human body, role of hepcidin, ferritin, transferrin. An infographic could be made.

In order for iron supplementation in clinical practice to have quality, it is important that review studies support the formation of a critical view. For this reason, it is important that not only the advantages, but the disadvantages of each supplement are presented. For example: information like this “and even a higher risk of colorectal carcinoma” should be better explained. How long to use? What's the dosage? What is the occurrence by age group? Did more than one researcher observe the same outcome?

The manuscript should be better organized between severe cases and mild cases.

The term bioavailability has been used since the beginning of the text, but its definition has not been described. Will readers confuse it with bioaccessibility?

The bioaccessibility of supplements should be mentioned.

In item 8- “Prevention of iron deficiency”, the amount of heme and non-heme iron in foods could be presented in the form of a table.

Please inform in the text the studies has been carried out in adults. For example, references 7, 11, 15 and 16.

Study 15 does not mention adolescents. Page 5 lines 164-165 “Therefore, despite the usual dosage in textbooks is 3-6 mg/kg/day of elemental iron in 2-3 administrations (60-120 mg/day in adolescents) [15]…”

Item 4- Diagnosis – Please show cutoff points according to sex and age for hematimetric and hepcidin indicators

Mention the relationship between iron and other nutrients (Silva AP, Pereira ADS, Simões BFT, Omena J, Cople-Rodrigues CDS, de Castro IRR, Citelli M. Association of vitamin A with anemia and serum hepcidin levels in children aged 6 to 59 mo. Nutrition. 2021 Nov-Dec;91-92:111463. doi: 10.1016/j.nut.2021.111463)

What is the effectiveness of daily and weekly iron supplementation in the prevention of anemia in infants?

Are all methods used to diagnose anemia validated? What is the validity of portable methods used to diagnose anemia? da Silva Pereira A, de Castro IRR, Bezerra FF, Nogueira Neto JF, da Silva ACF. Reproducibility and validity of portable haemoglobinometer for the diagnosis of anaemia in children under the age of 5 years. J Nutr Sci. 2020 Jan 20;9:e3. doi: 10.1017/jns.2019.43

Reviewer 2 Report

Dear Editor,

Moschero et.al. present a manuscript that is in fact excelent and thorough owerviev of iron deficiency anemia in childhood for practising pediatritian. There is not much that I found missing and there is not much that could be added to the existing text. Beside minor editorial issues (missing commas), I found the manuscript suitable for publication in present form.

Reviewer 3 Report

In this paper the authors review some aspects of iron deficiency anemia in children. The subject is interesting, however the manuscript is too generic and the conclusions do not clarify what message the authors wish to convey. Furthermore, in-depth analyses specifically regarding children are lacking. The authors should clarify what are the differences in children (no data for different age groups are shown/analysed) compared to adults. The role of hepcidin should be explained in more detail at the molecular level: ferroportin is not even mentioned, and there are many references explaining the molecular details of the role of protease TMPRSS6 in the regulation of hepcidin. Table 2 is a bit difficult to read and should be reformatted; also, it is unclear if the reported recommended dosages are suggested by the authors or if they are already established in the literature. 

Reviewer 4 Report

This review is probably a little too elementary as a publication in a scientific journal.  It is better suited to student readership for which it does quite well. Although one error is noted. The cord should be clamped after a delay to allow the cord blood to return to the neonate not immediately! This will increase the amount of blood retained.